# Stochastic sampling effects favor manual over digital contact tracing

Marco Mancastroppa [1,2], Claudio Castellano [3], Alessandro Vezzani [1,4] & Raffaella Burioni [1,2✉]

Isolation of symptomatic individuals, tracing and testing of their nonsymptomatic contacts are fundamental strategies for mitigating the current COVID-19 pandemic. The breaking of contagion chains relies on two complementary strategies: manual reconstruction of contacts based on interviews and a digital (app-based) privacy-preserving contact tracing. We compare their effectiveness using model parameters tailored to describe SARS-CoV-2 diffusion within the activity-driven model, a general empirically validated framework for network dynamics. We show that, even for equal probability of tracing a contact, manual tracing robustly performs better than the digital protocol, also taking into account the intrinsic delay and limited scalability of the manual procedure. This result is explained in terms of the stochastic sampling occurring during the case-by-case manual reconstruction of contacts, contrasted with the intrinsically prearranged nature of digital tracing, determined by the decision to adopt the app or not by each individual. The better performance of manual tracing is enhanced by heterogeneity in agent behavior: superspreaders not adopting the app are completely invisible to digital contact tracing, while they can be easily traced manually, due to their multiple contacts. We show that this intrinsic difference makes the manual procedure dominant in realistic hybrid protocols.

[1] Dipartimento di Scienze Matematiche, Fisiche e Informatiche, Università degli Studi di Parma, Parco Area delle Scienze, Parma, Italy. [2] INFN, Sezione di Milano Bicocca, Gruppo Collegato di Parma, Parco Area delle Scienze, Parma, Italy. [3] Istituto dei Sistemi Complessi (ISC-CNR), Via dei Taurini, Roma, Italy. [4] Istituto dei Materiali per l'Elettronica ed il Magnetismo (IMEM-CNR), Parco Area delle Scienze, Parma, Italy. ✉email: raffaella.burioni@unipr.it

The current COVID-19 pandemic is impacting daily life worldwide at an unprecedented scale. Among the features that have contributed to transform the emerging diffusion of SARS-CoV-2 coronavirus into such a global scale crisis, a prominent role is played by the high rates of virus transmission mediated by presymptomatic and asymptomatic individuals[1–6]. Given the absence of effective pharmaceutical interventions, this feature makes the mitigation of the pandemic a highly nontrivial task, that has been tackled with various strategies, none of them devoid of drawbacks.

Initially, governments resorted to very restrictive limitations of non strictly necessary activities (lock-downs) to curb the diffusion of the infection. Such measures turned out to be effective from an epidemiological point of view, but exceedingly costly in other respects, for their economic and social consequences[7,8]. Recently, such restrictive measures have been progressively lifted, and we now rely on other tools to contain the pandemic: social distancing, reinforced hygiene, and the use of individual protection devices. Along with these provisions, aimed at preventing single virus transmission events, another set of measures points at breaking contagion chains: the isolation of infected individuals (symptomatic or found via some testing), followed by the tracing of their contacts (contact tracing, CT), the testing of the latter, and the possible isolation of the infected[9–12].

This CT procedure has proven effective in the past in various contexts[13–16] but it comes, in its standard manual implementation, with important limitations[17]. It requires the set up of a physical infrastructure, needed to find infected individuals, interview them and reconstruct their contacts in a temporal window, call these contacts, convince them to get tested, and eventually isolated. Apart from the evident problems of practical feasibility and economic cost, the manual CT procedure intrinsically implies a delay between the moment an individual is found infected (and isolated) and the time her contacts are tested and possibly isolated. For an epidemic such as COVID-19, characterized by a rather long presymptomatic infectious stage and a high relevance of transmission by asymptomatics, the delay implied by manual CT risks to undermine the effectiveness of the whole procedure.

For this reason the additional strategy of a digital CT procedure, based in particular on the installation of apps on smartphones (app-based), has been proposed alongside manual CT[2]. The rationale is that proximity sensors installed on these ubiquitous devices allow the detection of contacts of epidemiological significance among individuals. When an individual is found infected, the app permits to instantaneously trace all contacts in the recent past, thus allowing for much quicker testing and isolation. A quantitative comparison between manual and digital CT applied to an epidemiological model describing COVID-19 diffusion suggested that already a delay of the order of 3 days completely spoils the ability of manual CT to prevent the initial exponential growth of the epidemic[2]. The conclusion was that only a digital CT avoiding this delay could be a viable strategy to control the current epidemic. The proposal for digital CT rapidly gained momentum, leading to the development of technical solutions[18–20] and to the deployment of app-based CT infrastructures in many countries[21].

Many works have scrutinized the actual validity of this solution and investigated the possible shortcomings of app-based CT, casting doubts over many of the assumptions underlying such strategy[22–24]: there are too few modern enough smartphones; Bluetooth-based proximity measurements are unreliable; co-location is not always a good proxy for epidemiological contact. The potential risks for privacy breaches have also been exposed. Other papers have tried to evaluate the impact of digital CT on the current COVID-19 epidemic, attempting to precisely

determine, by means of detailed data-driven epidemiological models, to what extent such a strategy is able to suppress virus diffusion[25–29]. A critical role is played by the fraction $f$ of individuals in a population that actually use the app. Fairly high values of $f$ (of the order of 60%) are required for the digital CT protocol to lead to global protection[2,9,23]. These values are in striking contrast with the low app adoption rates observed so far in most countries[30–32].

In this paper we take a different approach. We compare the effectiveness of the two CT protocols in exactly the same conditions, i.e., in the very same realistic epidemiological scenario, without making claims on their absolute performance, and we estimate their relative contribution in realistic hybrid protocols, where the two strategies are complementary. We consider a sensible epidemiological model incorporating all main ingredients of the current epidemic, with parameters tuned to values derived from empirical observations about COVID-19 spreading. Within this single framework we consider the impact of both manual and digital CT strategies, working in similar manner but with their own specific features: delayed isolation of contacts, limited scalability, and imperfect recall for the manual procedure; dependence on the predetermined app adoption decision for the digital CT.

The comparison reveals that even when the number of reconstructed contacts is the same, manual CT performs better than digital CT in practically all realistic cases. The manual protocol is more efficient in increasing the epidemic threshold (i.e., the value of the effective infection rate above which the infection spreads diffusely), in limiting the height of the epidemic peaks and in reducing the number of isolated individuals. This surprising result is due to the stochastic annealed nature of the manual CT procedure, in which each symptomatic node randomly recalls a fraction of her contacts, in contrast with the digital CT where the traced nodes belong deterministically to the prearranged quenched fraction of the population adopting the app. In the latter case, the individuals not adopting the app can never be reached by the CT protocol, while the entire population is potentially detectable through the stochastic sampling of the manual procedure. The better performance of manual CT is already evident in homogeneous populations and it is strongly enhanced in the presence of a heterogeneous distribution of contacts. Super-spreaders not adopting the app are invisible to the digital tracing while they are very likely to be detected by a manual tracing originating from one of their many contacts. In a realistic setup where both protocols are adopted simultaneously, manual CT leads to a considerable reduction of the transmission even considering delays and scalability, while the digital protocol produces a relevant contribution only for large adoption rates, as suggested by previous literature.

## Results
**Epidemic spreading on heterogeneous dynamical networks**. We consider an activity-driven network model with attractiveness, taking into account both the temporal dynamics of social contacts and the heterogeneity in the propensity to establish social ties[33–36]. Each susceptible node $S$ is assigned with an activity $a_S$ and an attractiveness parameter $b_S$, drawn from the joint distribution $\rho(a_S, b_S)$: the activation rate $a_S$ describes the Poissonian activation dynamics of the node; the attractiveness $b_S$ sets the probability $p_{b_S} \propto b_S$ for a node to be contacted by an active agent. At the beginning all nodes are disconnected and when a node activates it creates $m$ links with $m$ randomly selected nodes (hereafter we set $m = 1$); then all links are destroyed and the procedure is iterated. The functional form of $\rho(a_S, b_S)$ encodes the correlations between activity and attractiveness in a population

**a**

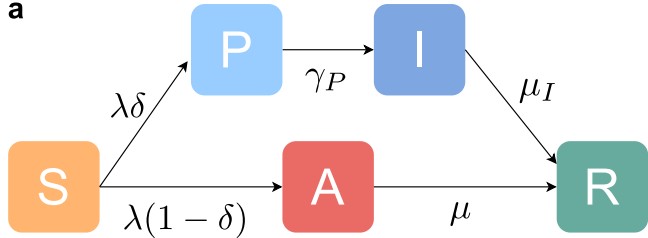

**b**

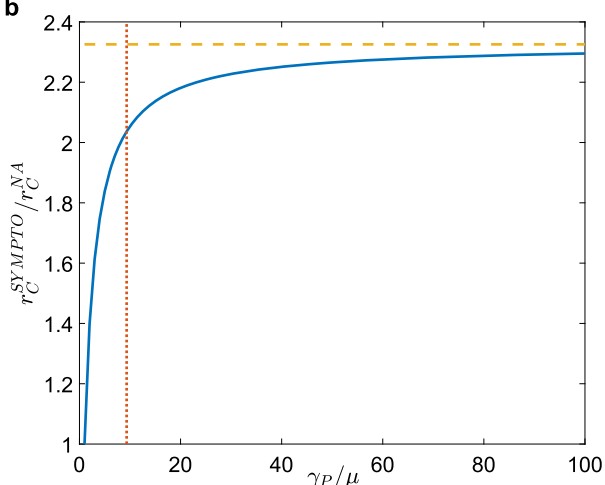

**Fig. 1 Epidemic model without contact tracing. a** Diagram of the compartmental epidemic model without CT. **b** We plot, as a function of $\gamma_P/\mu$, the ratio between the epidemic threshold $r_C^{SYMPTO}$ when symptomatic nodes are isolated and the epidemic threshold of the non-adaptive case $r_C^{NA}$. The horizontal dashed orange line indicates the value of $r_C^{SYMPTO}/r_C^{NA}$ for $\gamma_P = \infty$ (instantaneous onset of symptoms). The vertical red dash-dotted line indicates the value of $\gamma_P/\mu$ we consider in the rest of the paper. We set $\rho(a_S, b_S) = \rho_S(a_S)\delta(b_S - a_S)$. The curve does not depend on the specific form of $\rho_S(a_S)$.

with a given distribution of activity. It has been observed that several social systems feature positive correlations between activity and attractiveness and a broad power-law distribution of activity[33–39]:

$$\rho(a_S, b_S) \sim a_S^{-(\nu+1)}\delta(b_S - a_S) \qquad (1)$$

with $\nu$ typically ranging between 0.5 and 2.

On top of the activity-driven dynamics, we consider a compartmental epidemic model which includes the main phases of clinical progression of the SARS-CoV-2 infection[6,40–42], also applicable to other infectious diseases with asymptomatic and presymptomatic transmission. The model is composed by five compartments: $S$ susceptible, $P$ presymptomatic, $A$ infected asymptomatic, $I$ infected symptomatic, $R$ recovered. A contagion process (see Fig. 1a) occurs with probability $\lambda$ when a link is established between an infected (either $P$, $A$, and $I$) and a susceptible node $S$ (contact-driven transition): a node has probability $\delta$ to become presymptomatic after infection and probability $(1 - \delta)$ to become asymptomatic, thus $S \xrightarrow{\lambda\delta} P$ and $S \xrightarrow{\lambda(1-\delta)} A$. A presymptomatic node spontaneously develops symptoms with rate $\gamma_P = 1/\tau_P$, thus with a Poissonian process $P \xrightarrow{\gamma_P} I$; both asymptomatic and symptomatic nodes spontaneously recover respectively with rate $\mu = 1/\tau$ and with rate $\mu_I = \mu\gamma_P/(\gamma_P - \mu)$, so that the average infectious period for both symptomatic and asymptomatic is $\tau$. We neglect states of hospitalization and consider recovery without death: this choice does not affect the infection dynamics.

Adaptive behavior of populations exposed to epidemics can be modeled within the activity-driven network framework: infected nodes experience a reduction in activity, due to isolation or the appearance of symptoms; similarly, other individuals undertake self-protective behavior to reduce the probability of contact with an infected node, and this is modeled as a reduction in the attractiveness of infected nodes[43–45]. We assume that symptomatic infected nodes $I$ are immediately isolated $(a_I, b_I) = (0, 0)$, therefore not being able to infect anymore. On the contrary, we assume that recovered $R$, asymptomatic $A$, and presymptomatic $P$ individuals behave as when they were susceptible $(a_A, b_A) = (a_P, b_P) = (a_R, b_R) = (a_S, b_S)$. The adaptive behavior is implemented without affecting the activity of nodes which are not isolated[44].

The control parameter $r = \lambda/\mu$ is the effective infection rate, whose critical value $r_C$—the epidemic threshold—sets the transition point between the absorbing and the active phase of the epidemic. The increase in the value of $r_C$ is an indicator of the effectiveness of mitigation strategies. Within the adaptive activity-driven framework, the epidemic threshold can be calculated analytically via a mean-field approach (see Methods).

The effect of isolating symptomatic nodes as the only containment measure is shown in Fig. 1b. We compare the epidemic threshold $r_C$, obtained with the isolation of symptomatic nodes only, with the epidemic threshold $r_C^{NA}$ of the non-adaptive (NA) case, in which no containment measures are taken on infected individuals, as a function of $\gamma_P/\mu$ (see Methods for the explicit expression). In the case of instantaneous symptoms development ($\gamma_P/\mu \to \infty$), the threshold is increased by a factor of $1/(1 - \delta)$, while for smaller $\gamma_P/\mu$ the gain is reduced. For example, for $(1 - \delta) = 0.43$, that is 43% of asymptomatic individuals, $\tau_P = 1.5$ days and $\tau = 14$ days as observed for SARS-CoV-2 (see Methods for details on the parameters used in all figures), the threshold is doubled by the isolation of symptomatic nodes. This is the baseline reference for the evaluation of the performance of CT strategies.

**Manual and digital contact tracing protocols**. The CT protocols differ in their practical implementation as well in their exploration properties.

Manual tracing is performed by personnel who, through interviews, collects information, contacts individuals who may have been infected and arranges for testing. In manual CT, as soon as an individual develops symptoms (i.e., $P \to I$), her contacts in the previous $T_{CT}$ days are traced with recall probability $\epsilon(a_S)$, where $a_S$ is the activity of the symptomatic individual. A traced contact is tested and, if found in state $A$ (infected asymptomatic), is isolated ($a = b = 0$): the average time between the isolation of the symptomatic individual and the isolation of her asymptomatic infected contacts is $\tau_C$. Such delay can be quite large, due to the time required for the collection of the diary, the execution of the diagnostic test and the subsequent isolation[2,17]. Moreover, the manual protocol depends on $\epsilon(a_S)$, which takes into account the limited resources allocated for tracing and the limited memory/knowledge of symptomatic individuals in reconstructing their contacts. Low activity nodes make few contacts over time and a fraction of their contacts will be traced; on the other hand high activity nodes will only remember a finite number of their contacts so that, also because of limitations of the tracing capacity, we expect that at most a number $k_c$ of contacts can be traced[46,47]. This translates into the limited scalability property:

$$\epsilon(a_S) = \begin{cases} \epsilon^*, & \text{if } a_S \leq a^* \\ \epsilon^*\dfrac{a^*}{a_S} = \dfrac{k_c}{2T_{CT}a_S}, & \text{if } a_S > a^* \end{cases} \qquad (2)$$

where $a^* = k_c/2T_{CT}\epsilon^*$. Manual CT also suffers from a global

scalability limitation. Indeed in the active phase with a large number of infected individuals, the tracing system could be ineffective due to the excessive number of contacts to be followed. However, here we focus on the epidemic threshold and we evaluate the features of the CT procedure that keep the epidemic spreading under control before widespread diffusion occurs.

Digital CT is based on the download of an app which allows the tracing of close contacts equipped with the same app. We assume that each of the individuals has a probability $f$ to download the app before the epidemic starts. As soon as an individual develops symptoms (i.e., $P \to I$), if she downloaded the app, her contacts are traced only if they downloaded the app as well. A traced contact is tested and, if found in state $A$ (infected asymptomatic), is isolated ($a = b = 0$). The time passing between the isolation of a symptomatic individual and the isolation of her asymptomatic infected contacts is taken to be 0, thus assuming an idealized scenario of instantaneous notification and isolation. We finally consider a more realistic scenario where the two procedures are combined into a hybrid protocol in which digital CT supports manual CT, potentially reaching individuals not traced manually[19,25,29]. See Methods for details on the implementation of the CT protocols.

**Stochastic vs. prearranged sampling.** We first compare the manual and digital CT protocols in the case of a population with homogeneous activity and attractiveness, $\rho(a_S, b_S) = \delta(a_S - a)\delta(b_S - b)$, without delay even in manual CT, i.e., $\tau_C = 0$. We set $\epsilon = f^2$ so that the probability that a single contact is traced in the two protocols is the same. However, it should be emphasized that typical values of $f^2$ range between 0.01 and 0.1 in many countries[30–32], while $\epsilon$ is usually larger ($\approx 0.3-0.5$), since typically 30−50% of contacts are close and occur at home, at work, or at school and thus are easily traceable[48,49]. An exact analytical estimate of the epidemic threshold is obtained through a linear stability analysis around the absorbing state (see Methods for explicit expressions and Supplementary Method 1 for detailed derivation): in Fig. 2a we compare the threshold for manual and app-based CT, compared to the NA case, for realistic COVID-19 parameters. Both protocols feature the same epidemic threshold when $\epsilon = f = 0$ and $\epsilon = f = 1$: indeed the former corresponds to the isolation of symptomatic individuals only, without CT, while the latter limit corresponds to the case in which all contacts are traced. For intermediates values of $\epsilon = f^2$, manual tracing is strongly and surprisingly more effective than digital tracing, as it increases significantly the epidemic threshold, compared to the app-based protocol. Since we are considering no heterogeneity or delays, the difference is due only to sampling effects in the CT dynamics. In practice, in app-based CT the population to be tested is prearranged, based on whether or not the app was downloaded before the outbreak started. On the contrary, manual CT performs a stochastic sampling of the population: the random exploration can potentially reach the entire population, since anyone who has come in contact with a symptomatic node can be traced. The simplest example of the difference in tracing multiple infections processes in the two protocols is illustrated in Fig. 2b.

**Effects of heterogeneous activity.** We now consider a heterogeneous activity distribution, as observed in several human systems, and we consider a positive activity-attractiveness correlation, as defined in Eq. (1)[33–35,37–39]. The power-law distribution for the activity implies the presence of hubs with high activity and high attractiveness. We investigate the pure effect of heterogeneities in CT[12,27] setting now $\epsilon(a_S) = \epsilon, \forall a_S$ and not considering delays in manual CT, $\tau_C = 0$. We perform again a mean-field approach, obtaining an analytical closed form for the

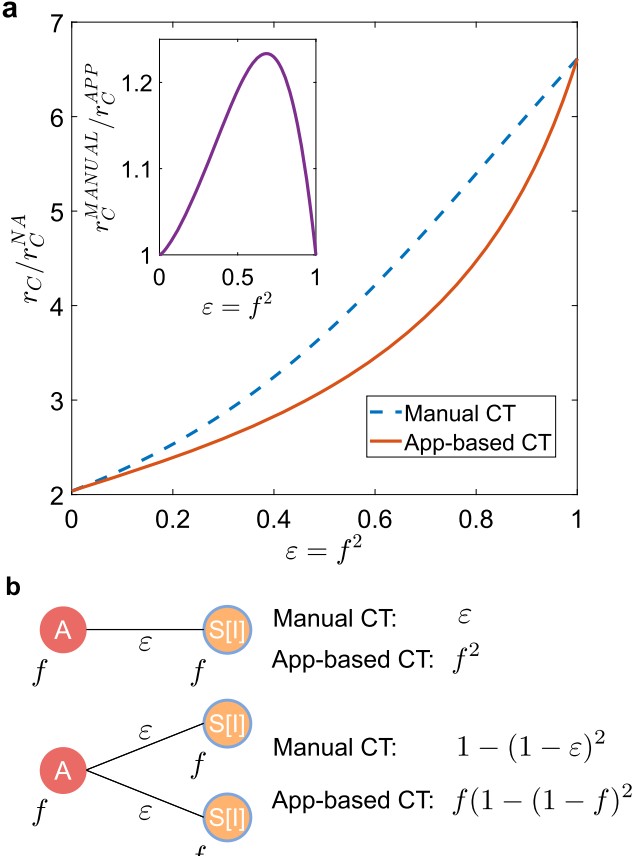

**Fig. 2 Stochastic vs. prearranged sampling in contact tracing. a** Plot, as a function of $\epsilon = f^2$, of the ratio between the epidemic threshold $r_C$ in the presence of CT protocols and the epidemic threshold of the non-adaptive case $r_C^{NA}$. In the inset we plot the ratio between the epidemic threshold of the manual CT $r_C^{MANUAL}$ and that of the app-based CT $r_C^{APP}$, as a function of $\epsilon = f^2$. We consider homogeneous activity and attractiveness, setting $\rho(a_S, b_S) = \delta(a_S - a)\delta(b_S - b)$. **b** If an asymptomatic node $A$ infects a single susceptible node $S$ (which subsequently becomes symptomatic, $I$), the infector is traced with probability $\epsilon$ in the manual CT and with probability $f^2$ in the digital CT. Thus the probability is the same if we impose $f^2 = \epsilon$. However, if an asymptomatic node infects two susceptible nodes (which subsequently become symptomatic), the probability of tracing the infector with the manual protocol is $1 - (1-\epsilon)^2 = 2f^2 - f^4$ (still considering $\epsilon = f^2$). This value is always larger than that of the digital protocol $f(1-(1-f)^2) = 2f^2 - f^3$.

epidemic threshold (see Methods and Supplementary Method 1): in Fig. 3 we compare the epidemic threshold with the two protocols as a function of the exponent $\nu$ of the activity distribution, for realistic parameters and setting an average activity $\overline{a_S} = 6.7$ days$^{-1}$[43,48]. Both protocols are more effective in heterogeneous populations, that is at $\nu \sim 1-1.5$. Note that due to the cut-offs and the constraint on the average, the fluctuations of the activity distribution are maximum for $\nu = 1$, and the epidemic thresholds depend both on activity fluctuations and higher order moments of $\rho_S(a_S)$ (see Methods and Supplementary Method 1 for details). However heterogeneity greatly amplifies stochastic effects, further increasing the advantage of the manual tracing over the app-based prearranged protocol. Indeed, in heterogeneous populations nodes with high activity and attractiveness (superspreaders) drive and sustain the spread of the epidemic. Manual CT is far more effective in identifying and isolating them than app-based CT: in digital CT, hubs which have not downloaded

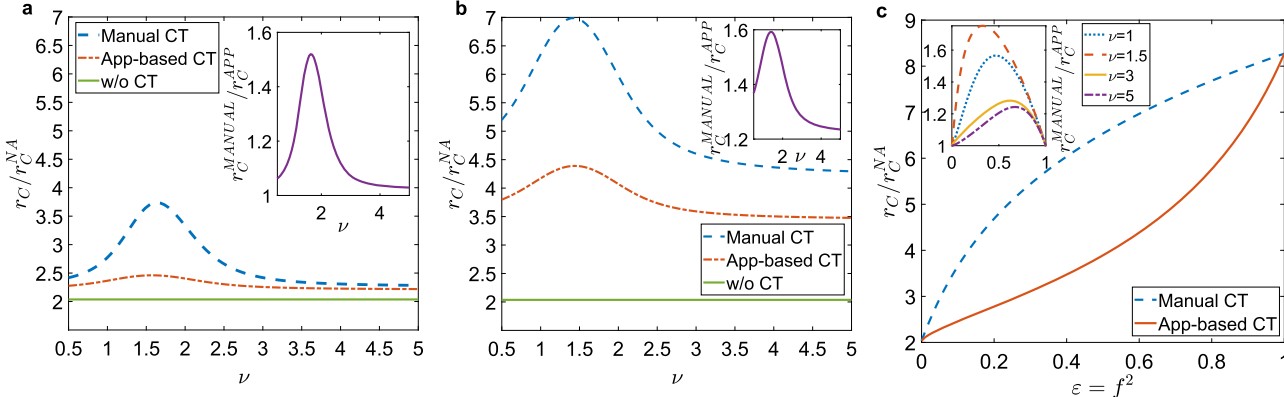

**Fig. 3 Effects of heterogeneity. a** Plot, as a function of the exponent $\nu$, of the ratio between the epidemic threshold $r_C$ in the presence of CT protocols and the epidemic threshold of the non-adaptive case $r_C^{NA}$. We set $\epsilon = f^2 = 0.1$ ($f \approx 0.316$). In the inset we plot the ratio between the epidemic threshold of the manual CT $r_C^{MANUAL}$ and that of the app-based CT $r_C^{APP}$, as a function of $\nu$. **b** Same plot as **a** with $\epsilon = f^2 = 0.6$ ($f \approx 0.775$). **c** The ratio $r_C/r_C^{NA}$ is plotted as a function of $\epsilon = f^2$ for both CT protocols, with $\nu = 1.5$. In the inset we plot the ratio $r_C^{MANUAL}/r_C^{APP}$ as a function of $\epsilon = f^2$ for several $\nu$ values. In all panels the distribution $\rho(a_S, b_S)$ is given by Eq. (1).

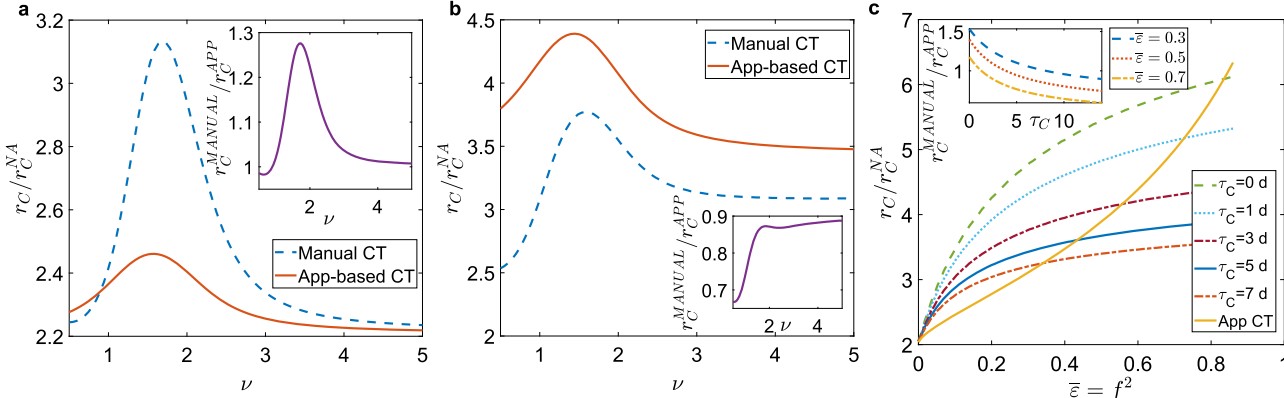

**Fig. 4 Effects of limited scalability and isolation delay in manual contact tracing. a** Plot, as a function of the exponent $\nu$, of the ratio between the epidemic threshold $r_C$ in the presence of CT protocols and the epidemic threshold of the non-adaptive case $r_C^{NA}$, with limited scalability and delay in manual CT. We set $\bar{\epsilon} = f^2 = 0.1$ ($f \approx 0.316$), $\tau_C = 3$ days. In the insets we plot the ratio between the epidemic threshold of the manual CT $r_C^{MANUAL}$ and that of the app-based CT $r_C^{APP}$, as a function of $\nu$. **b** Same plot as **a** with $\bar{\epsilon} = f^2 = 0.6$ ($f \approx 0.775$), $\tau_C = 5$ days. **c** Plot of the ratio $r_C/r_C^{NA}$ as a function of $\bar{\epsilon} = f^2$ for both CT protocols, setting $\nu = 1.5$ and for several values of $\tau_C$. In the inset we plot the ratio $r_C^{MANUAL}/r_C^{APP}$ as a function of $\tau_C$ for several $\bar{\epsilon}$ values. In all panels the distribution $\rho(a_S, b_S)$ is given by Eq. (1).

the app will never be traced, despite the high number of their contacts. On the contrary, manual CT is very effective in tracing super-spreaders, because they are engaged in many contacts and are traced very effectively by stochastic exploration.

**Limited scalability and delay in manual CT**. We now consider some features of manual CT that can reduce its effectiveness: the limited scalability of the tracing capacity[46,47] and the delays in CT and isolation[2,23,28]. We set $\epsilon(a_S)$ as in Eq. (2) and consider a large delay $\tau_C = 3$ days[2] in manual CT. In Fig. 4 we compare the epidemic threshold for the two CT protocols, setting equal probabilities of tracing a contact $\bar{\epsilon} = f^2$, where $\bar{\epsilon} = \int da_S \epsilon(a_S) \rho_S(a_S)$. For small values of $\bar{\epsilon}$ (note that this however corresponds to a quite large adoption rate $f = \sqrt{\bar{\epsilon}} \approx 0.316$) manual CT is still more effective (see Fig. 4a): the delay in isolation and the limited scalability are not able to significantly reduce the advantage provided by the stochastic exploration of contacts. Figure 4b shows that digital CT can become more effective than manual CT, but this occurs only for very large values of $f^2$ and $\tau_C$. This indicates that for realistic settings, the advantage of the manual protocol over the app-based protocol is robust even including delays and limited scalability. Figure 4c

further illustrates for which (unrealistically large) values of $f^2$ and $\tau_C$, digital CT outperforms manual CT. Note that realistic values of $f$ correspond to $f^2$ at most of the order of 0.1. Hence, an extremely high adoption of the app is necessary in order to obtain an effective advantage of the digital CT.

**Manual and app-based CT in the epidemic phase**. We now explore with numerical simulations (see Methods, Supplementary Method 2) the effects of manual and digital CT protocols in the active phase of the epidemic. We consider an optimistic value of $f = 0.316$, setting $\bar{\epsilon} = f^2 = 0.1$ that is a very low value for the recall probability, and we consider the system above the epidemic threshold $r > r_c$, in the conditions of Fig. 4a. Figure 5 shows that the infection peak with manual tracing is lower than the app-based one. Moreover, in the manual CT the duration of the epidemic is reduced: this strongly impacts on the final epidemic size, which is about half of the one observed in the app-based CT. We also plot the temporal evolution of the average activity of the system $\langle a(t) \rangle$ and of the fraction of isolated nodes Iso(t) (see inset). In general, the average activity $\langle a(t) \rangle$ features a minimum, however its value remains very large (about 98% of the case without any tracing measure). This implies that both protocols do

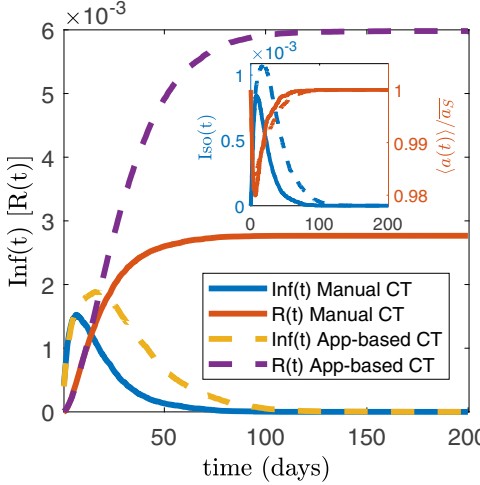

**Fig. 5 Effects of manual and digital contact tracing on the epidemic active phase.** We plot the temporal evolution of the fraction of infected nodes Inf($t$), i.e., infected asymptomatic and infected symptomatic, and of the fraction of removed nodes R($t$), for both manual and digital CT. In the inset we plot the temporal evolution of the fraction of isolated nodes Iso($t$) (right y-axis) and of the average activity of the population $\langle a(t) \rangle$ (left y-axis), normalized with $\overline{a_S}$. All curves are averaged on several realizations of the disorder and of the temporal evolution. We set $\overline{\epsilon} = f^2 = 0.1$, $\tau_C = 3$ days, $r/r_C^{NA} = 3.1$ and $N = 5 \times 10^3$. The distribution $\rho(a_S, b_S)$ is given by Eq. (1), with $\nu = 1.5$. The curves for digital and manual CT are averaged respectively over 554 and 681 realizations and the errors, evaluated through the standard deviation, are smaller or comparable with the curves thickness.

not disrupt the functionality of the system. Interestingly, the fraction of isolated nodes is coherent with the infection peak, and in particular it is lower when the infection peak is lowered. This means that the most effective procedure, i.e., manual CT for realistic values of the parameters, not only lowers the infection peak but it is also able to isolate a smaller number of nodes, a key feature of any effective CT strategy.

**Robustness**. In the Supplementary Notes we show that the advantage of the manual CT is robust with respect to the relaxation of several assumptions and to changes of parameters. In particular, we consider the case where all nodes have equal attractiveness $\rho(a_S, b_S) = \rho_S(a_S)\delta(b_S - b)$, we take into account very long delays $\tau_C$ and we change the maximum number of traceable contacts $k_c$. We also show that the advantage of manual CT in the presence of heterogeneous activity is robust, if one takes into account that contacts belonging to the close social circle of the index case (same household) are always traced and isolated manually, also in the digital protocol. This can be verified by using a hybrid procedure with a fixed small number of contacts always traced (see Supplementary Notes).

**Hybrid CT protocols: manual plus digital CT**. We now consider the implementation of a realistic hybrid CT protocol with manual and digital CT working at the same time. We assume that each individual has downloaded the app with probability $f$ before the epidemic starts. As soon as an individual with activity $a_S$ develops symptoms (i.e., $P \rightarrow I$), CT is activated with reference to the previous $T_{CT}$ days: if the infected individual and also her contact downloaded the app, the contacted agent is immediately traced and isolated with no delay. Otherwise, the contact can only be traced manually, with probability $\epsilon(a_S)$ (as in Eq. (2)) and, if found in state $A$, the contact is isolated with a delay $\tau_C$. We set a realistically large delay $\tau_C = 3$ days.

The threshold in the hybrid case can be determined through the solution of a complex set of equations, that we solve numerically in Supplementary Method 1. In the presence of heterogeneous activity (i.e., super-spreaders), starting form a purely digital protocol (Fig. 6b, c) the addition of manual CT rapidly increases the threshold, leading to an improvement by a large factor (about 80%), for realistic values $\overline{\epsilon} \gtrsim 0.3$. Instead, starting from a realistic manual CT setup $\overline{\epsilon} \gtrsim 0.3$, a significant improvement (i.e., a 50% of threshold increase) is obtained by implementing a digital CT only if $f \gtrsim 0.60-0.75$ (Fig. 6a, c). These values are consistent with other results in the literature[2,9,23]; however, in most countries, the app adoption rates do not reach these values[30–32].

### Discussion

Our results indicate that manual CT, despite its drawbacks, can be an efficient protocol in heterogeneous populations, more efficient than its digital counterpart, due to its specific sampling properties. This conclusion is robust with respect to variations in several model assumptions, including correlations between activity and attractiveness or the limited scalability of the manual CT protocol. However, epidemic propagation and strategies to mitigate it are very complex processes and several of their features have been left out from our modeling scheme. Some of these features (the possibility that isolation is non complete, that some individuals do not report symptoms, or the existence of testing campaigns detecting infected nonsymptomatic individuals) act similarly on both types of CT and hence do not modify the relative performance. Other more realistic features (the presence of delays even in digital CT and the existence of additional sources of heterogeneity in viral shedding[50], recovery rates[51], and activity temporal patterns[52]) would even reduce the relative performance of digital CT.

Our results put forward several directions to increase the effectiveness of tracing. An important aspect is the correlation between app adoption rates and activity of nodes: our analysis in the Supplementary Notes shows that, as expected, in heterogeneous populations a positive correlation strongly increases the success of digital CT. This is a direction that could be pursued in campaigns aimed at driving app adoption among potential super-spreaders. However, this represents a challenge for policy makers. Evidence is currently emerging that those who download the app are individuals who adopt very cautious behavior, i.e., $f$ and $a_S$ are typically anticorrelated[53,54]. Another road that could enhance the efficacy of CT protocols is to follow chains of transmissions along multiple steps, so that when a traced contact is found infected also her contacts are reconstructed and tested. This additional step improves the overall effectiveness of CT protocols, but also increases the delay associated to the manual procedure with respect to the digital one. Digital CT allows in principle to extend the tracing procedure to an arbitrary number of steps, however, strong concerns related to privacy issues[55] make this path difficult to follow.

In summary, even if additional features of CT can be considered, the weakness of digital CT, originated by the nature of the sampling of contacts and worsened by heterogeneities, seems to be an intrinsic unavoidable property of the procedure. The manual CT protocol, with its higher intrinsic stochasticity, does not suffer from this problem and samples contacts effectively, especially in realistic heterogeneous populations: thus, digital CT cannot be considered simply as a cheaper and more rapid way of implementing standard CT and should only be considered in combination with manual protocols. Manual CT must necessarily play an important role in any strategy to mitigate the current pandemic. Considerations about costs and practical feasibility of

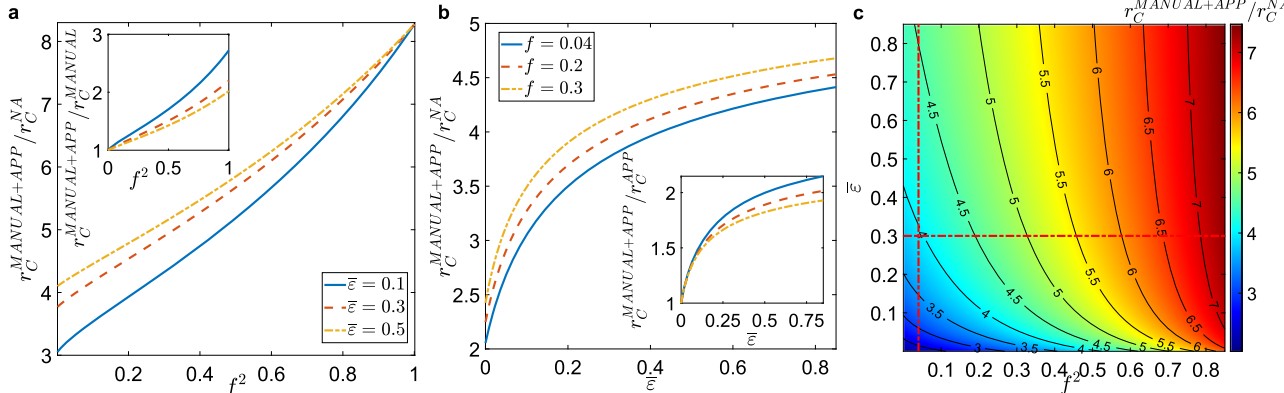

**Fig. 6 Effects of hybrid contact tracing protocol. a** Plot, as a function of $\hat{f}^2$, of the ratio between the epidemic threshold in the presence of a hybrid CT protocol, $r_C^{MANUAL+APP}$, and the epidemic threshold of the non-adaptive case, $r_C^{NA}$, for realistic values of $\bar{\epsilon}$ (see legend). In the inset we plot, as a function of $\hat{f}^2$, the ratio between $r_C^{MANUAL+APP}$ and the epidemic threshold when only manual CT is implemented, $r_C^{MANUAL}$. **b** Plot of the ratio $r_C^{MANUAL+APP}/r_C^{NA}$ as a function of $\bar{\epsilon}$, for realistic app adoption levels $f$ (see legend). In the inset we plot, as a function of $\bar{\epsilon}$, the ratio between $r_C^{MANUAL+APP}$ and the epidemic threshold when only app-based CT is implemented, $r_C^{APP}$. **c** The ratio $r_C^{MANUAL+APP}/r_C^{NA}$ is plotted as a function of $\hat{f}^2$ and $\bar{\epsilon}$, through a heat map: the red dash-dotted lines are plotted for $\bar{\epsilon} = 0.3$ and $f = 0.2$, i.e., they correspond to the red dashed curves of panel **a** and **b**. In all panels the distribution $\rho(a_S, b_S)$ is given by Eq. (1), the manual CT features a delay of $\tau_C = 3$ days and limited scalability is considered.

the two approaches (which have not been taken into account here) suggest that a careful integration of the two protocols may be the key for more effective mitigation strategies. As suggested in the last part of our results, an optimal set up should include both procedures, adopted simultaneously. In this respect, the availability of detailed data about the rates of app adoption in various population groups (and correlations with age or activity levels), as well as more precise estimates of other parameters, such as the recall probability, would be highly beneficial. The design of optimal hybrid CT protocols, including decisions on how to allocate resources and how to target recommendations for app adoption, is a very promising direction for future work.

## Methods

**Mean-field equations and implementation of CT protocols.** We consider an activity-attractiveness based mean-field approach, dividing the population into classes of nodes with the same activity $a_S$ and attractiveness $b_S$ and treating them as if they were statistically equivalent. For each class $(a_S, b_S)$ we consider the probability that at time $t$ a node is in one of the epidemic compartments. For arbitrary $\rho(a_S, b_S)$ distribution and arbitrary functional form of $f(a_S)$ and $\epsilon(a_S)$ we build the mean-field equations which describe the temporal evolution of the network, the epidemic spreading and the adaptive behavior due to isolation and CT. In particular, in order to model the manual and app-based CT we introduce two further compartments: $T$ traced asymptomatic and $Q$ isolated asymptomatic. An asymptomatic individual became traced $T$ when infected by a presymptomatic node or when it infects a susceptible node that eventually develops symptoms. In the manual case the tracing is effective with probability $\epsilon(a)$. In the app-based case, tracing occurs only if both nodes involved in the contact downloaded the app. A traced node is still infective $(a_T, b_T) = (a_S, b_S)$ and with rate $\gamma_A = 1/\tau_A \gg \mu$ it is quarantined, $T \xrightarrow{\gamma_A} Q$; while a quarantined node is no more infective since $(a_Q, b_Q) = (0, 0)$. In order to take into account the delay in the manual CT we set: $\tau_A = \tau_C + \tau_P$ for the manual case and $\tau_A = \tau_P$ for the app-based process. See Supplementary Method 1 for the detailed equations of manual, digital, and hybrid CT protocols.

**Epidemic thresholds.** We perform a linear stability analysis of the mean-field equations around the absorbing state obtaining the conditions for its stability and then the epidemic threshold $r_C$ (see Supplementary Method 1 for details). For the NA case, when infected individuals do not modify their behavior, the threshold is equal to the one obtained in refs. [35,44]. Indeed, setting $\rho(a_S, b_S) = \rho_S(a_S)\delta(b_S - a_S)$ we obtain:

$$r_C^{NA} = \frac{\overline{a_S}}{2\overline{a_S^2}} \tag{3}$$

If only symptomatic nodes are isolated as soon as they develop symptoms, we obtain, setting $\rho(a_S, b_S) = \rho_S(a_S)\delta(b_S - a_S)$:

$$r_C^{SYMPTO} = r_C^{NA} \frac{\frac{\gamma_P}{\mu}}{\delta + (1-\delta)\frac{\gamma_P}{\mu}} \tag{4}$$

For the case with CT on homogeneous population, we set $\rho(a_S, b_S) = \delta(a_S - a)\delta(b_S - b)$, $\epsilon(a_S) = \epsilon$ and no delay in manual CT $\tau_C = 0$. We obtain the epidemic threshold for both the manual and digital CT:

$$r_C^{MANUAL} = \frac{2r_C^{NA}\frac{\gamma_P}{\mu}}{\delta + (1-\delta-\epsilon\delta)\frac{\gamma_P}{\mu} + \sqrt{(\delta + (1-\delta-\epsilon\delta)\frac{\gamma_P}{\mu})^2 + 4\delta^2\epsilon\frac{\gamma_P}{\mu}}} \tag{5}$$

$$r_C^{APP} = \frac{2r_C^{NA}\frac{\gamma_P}{\mu}}{\delta + (1-\delta-f\delta)\frac{\gamma_P}{\mu} + \sqrt{(\delta + (1-\delta-f\delta)\frac{\gamma_P}{\mu})^2 + 4\delta f\frac{\gamma_P}{\mu}(\delta + \frac{\gamma_P}{\mu}(1-f)(1-\delta))}} \tag{6}$$

The more general case of populations with arbitrary distribution $\rho(a_S, b_S)$, arbitrary delays $\tau_C$ and general form of $\epsilon(a_S)$ and $f(a_S)$, is reported in Supplementary Method 1. In Supplementary Method 1 we also report the derivation of the epidemic threshold for the hybrid protocol. Its values are derived from the stability conditions of a complex set of 22 differential equations, that we can solve numerically.

**Model parameters.** Figures present results where one of the model parameters is varied and all the others are fixed. Here we report the parameter values used throughout the paper, unless specified otherwise. They are tailored to describe the current COVID-19 pandemic.

The fraction of infected individuals who develop symptoms is $\delta = 0.57$[1]. The time after which a presymptomatic individual spontaneously develops symptoms is $\tau_P = 1.5$ days[6,41,49], while infected individuals recover on average after $\tau = 14$ days[42,50]. The time window over which contacts are reconstructed is $T_{CT} = 14$ days[49]: it is fixed equal to $\tau$ to track both nodes infected by the index case in the presymptomatic phase (forward CT) and the primary case who infected the index case (backward CT)[12]. The maximum number of contacts engaged in $T_{CT}$ by a single individual that can be reconstructed with the manual CT procedure is $k_C = 130$, according to reasonable estimates of the number of contacts manually traced for very active individuals[49]. Moreover, fixing $k_C = 130$ and for realistic $\bar{\epsilon} \sim 0.1 - 0.5$[48], the average number of traced contacts for each index case is approximately $10-60$, consistently with reported data on manual CT and with estimates for resources allocation[46,49,56,57] (see Supplementary Notes for the distribution $P(k_T)$ of contacts $k_T$ traced manually by each index case). The activity-driven network parameters are fixed so that the average value of the activity is always the same, i.e., $\overline{a_S} = 6.7$ days$^{-1}$[43,48]. In particular for a power-law distribution $\rho_S(a_S) \sim a_S^{-(\nu+1)}$, the values of $a_S$ are constrained between a minimum and a maximum value ($a_m < a_S < a_M$). We keep $a_M = 10^3 a_m$ and then we tune $a_m$ to set $\overline{a_S}$.

**Numerical simulations.** We perform numerical simulations of the epidemic model on the adaptive activity-driven network: the network dynamics and epidemic spreading are implemented by a continuous time Gillespie-like algorithm. We consider an activity-driven network of $N$ nodes. The results are averaged over several realizations of the disorder and of the dynamical evolution, so that the error on the infection peak height is lower than 6%. The initial conditions are imposed by infecting the node with the highest activity $a_S$ and the CT protocols are

immediately adopted. A detailed description of the simulations is reported in Supplementary Method 2.

## Data availability

The data that support this study are available from the corresponding author upon reasonable request.

## Code availability

The simulation codes that support this study are available from the corresponding author upon reasonable request.

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

## Author contributions

M.M., C.C., A.V., and R.B. designed research, M.M., C.C., A.V., and R.B. performed research, M.M. analyzed data, M.M., C.C., A.V., and R.B. wrote the paper.

## Competing interests

The authors declare no competing interests.
