## [Peer Review File · Nature Communications]

Reviewer #1 (Remarks to the Author):

The article compares manual contact tracing with the approach based on exposure notification apps. The core message is that if an average probability of tracing a contact is assumed to be the same between manual and digital approach - even considering the delays inherent in the manual method - manual tracing is more efficient. The main reason behind this effect is that in the digital approach it is pre-determined who can and cannot be identified as a contact (by the virtue of having or not having a notification app installed). A super-spreader without an app will not be identified digitally, but is more likely to be identified manually because of their many contacts.

The paper points out an important shortcoming of digital contact tracing - the potential to miss superspreaders. As we learn more about the dynamics of spread, with a minority of carriers infecting the vast majority of susceptible individuals, this shortcoming proves to be more important than initially thought. Just because of this observation, I believe the paper constitutes a very important voice in the discussion. The presented results indicating comparatively high efficiency of manual tracing even with long delays are surprising and worth noting, since the slow operation is one of the main weaknesses pointed out against manual tracing. The paper is timely, as many countries still have not deployed digital tracing in any form while still struggling with implementing manual tracing. It is also written very clearly, most of the assumptions are well argued (apart from assuming that individuals with symptoms isolate immediately, but I don't believe it changes the outcomes of the comparison), and further examined for robustness in supplementary information.

I am not qualified to comment on the correctness of the results regarding the calculation of epidemic thresholds, but the methods appear to be derived from established papers. I can however comment on framing of the research question. The main problem with the paper is that it presents the two approaches to contact tracing as a dichotomous choice. However, there is an apparent consensus, including even the creators of the DP3T privacy preserving digital approach, that digital exposure notification systems should only supplement manual tracing, not replace it. Most countries have manual tracing systems in operation and their question is whether it makes sense to adopt an app-based system in addition to the traditional approach. If the authors were to make a truly important contribution to the current discussion, they should acknowledge this current situation, and perform the simulations when both systems are employed concurrently. Finally, I recommend that the authors refer in their discussion (or at least consider, as I am not sure about the policy around citing to the initial reports showing that app adoption appears to be reversely correlated with activity: people who install the app, tend to be more health conscious, work from home, and employ a range of other means to slow down the pandemic [2, 3]. That effect speaks even more strongly to the authors' point.

I believe that pointing out the super-spreader weakness of digital contact tracing is crucial and the paper deserves publishing mostly as is. However, I would recommend that the authors rephrase the manuscript so as it does not present the choice between the two approaches as either or. Ideally, the authors would present the results of simulating both systems at work simultaneously, but the paper can also be published without it.

[1] Digital Proximity Tracing in the COVID-19 Pandemic on Empirical Contact Networks, G. Cencetti et al (2020)

[2] Are COVID-19 proximity tracing apps working under real-world conditions? Indicator development and assessment of drivers for app (non-)use, V. Wyl et al (2020)

[3] Predicting public take-up of digital contact tracing during the COVID-19 crisis: Results of a national survey, Saw et al (2020)

Reviewer #2 (Remarks to the Author):

The article presents an extremely policy relevant study with fascinating results. While I commend the effort, I have a number of remarks that I hope the authors could address to clarify the scope and applicability (and possibly validity) of their study.

The authors indicate realism as one of the motivations of their work, my main concern is with some of the assumptions of the study, which may end up restricting its realism and applicability.

A crucial implementation assumption when comparing the two CT strategies seems to be that all contact tracing relies exclusively on the strategy tested. While it is realistic to assume that, in the manual contact tracing (MCT) scenario, all tracing relies on recollection on the part of the individual, it is not as realistic to assume that all contact tracing in the digital contact tracing (DCT) scenario relies to the adoption of the app. E.g: if I test positive to the virus, my wife and children will know and isolate, even if they didn't install the app.

It would be beneficial to the realism (and policy applicability) of the study if the authors assumed that a certain proportion (or a fixed number) of an individual's contacts are always traced, regardless of CT strategy, assuming that contacts within the household, or otherwise very proximal, can be traced without the need for a formal CT process.

Certain assumptions around the number of contacts being traced seem to implicitly favour the manual strategy. In particular, a maximum of $k_C=130$ seems overwhelmingly large, even according to the literature that the authors themselves refer to. Ref.52 (page 6) mentions 10-30 as the interval of contacts normally manually traceable for each individual. The values tested in the sensitivity analysis also appear to overestimate this number. Traceable or not, 130 contacts might be an extreme case for an individual with high a and b in the power-law network. I'd like the authors to add some detail on the contact patterns emerging from their model. A simple plot of the distribution of contacts and/or infection events could be useful to assess the realism of the patterns generated.

The limited scalability assumption also seems very lenient on the MCT strategy. The only scalability issue implemented is a limitation to the recollection ability of individuals. However we know that, unfortunately, when the infection rate becomes high enough, MCT systems with limited manpower are likely to collapse, resulting in a failure to interview a proportion of infected agents altogether. DCT should (at least in principle) escape this further limitation as they are less reliant on manpower.

I believe that, in a revised version of the paper, the authors should, at least, add this remark to the limitations of the model. Or they could try and implement a more realistic scalability limitation by not only assuming a lower k_C , but also that some nodes may not return any contact (or very few, just the closest) to the MCT system, especially at the peak of the epidemic.

One final remark. In the Introduction the authors seem to imply that DCT is an alternative to MCT. This is not the case: where DCT is implemented it is alongside MCT, the two strategies are seen as complementary, not alternative. After all, as the authors correctly state, generally around 50% of contacts can be easily traced. DCT exists for the other 50%.

There's a mistype in the Fig.2 caption, where the exponent of the first term of the last equation should be outside the parenthesis. The calculation is correct nevertheless.

Reviewer #3 (Remarks to the Author):

The authors investigated the effectiveness of manual and digital contact tracing using an activity-driven model with parameters tailored to COVID-19.

The authors showed that, although digital contact tracing can be performed instantaneously at scales, digital contact tracing is less effective than manual contact tracing.

This is because digital contact tracing cannot trace infections from or to people without a tracing app while manual tracing can, calling a necessity of manual contact tracing to mitigate the current pandemic.

The manuscript is well written and clear in most of its passages, and I have some minor comments and questions for the authors which I think can improve readability.

- What are the noteworthy results?

Manual contact tracing can be more effective than digital contact tracing even when digital contact

tracing is performed instantaneously at scales.

- Will the work be of significance to the field and related fields? How does it compare to the established literature? If the work is not original, please provide relevant references.

Yes. The authors clearly position their work in the literature. Specifically, related studies investigated the effectiveness of digital contact tracing based on simulations and case reports, calling that app adoption rate plays a crucial role.

Essentially, the authors approach the same question but take a different approach, where they compare the effectiveness of manual and digital contact tracing.

- Does the work support the conclusions and claims, or is additional evidence needed?

One issue is that they assume that manual tracing can trace at most k_c contacts per each infected individual. This is indeed not true for manual contact tracing; as the number of new cases continues to grow, one may be compelled to spare less time and resources per individual. Therefore, the number of traceable contacts per individual would decrease as the new cases grow. Therefore, their claim is limited to the early stage of epidemic spreading, which is neither discussed nor stated in the manuscript.

- Are there any flaws in the data analysis, interpretation, and conclusions? - Do these prohibit the publication or require revision?

No.

- Is the methodology sound? Does the work meet the expected standards in your field?

I suggest the authors to show the variations (e.g., standard error or confidence interval) in the figures.

- Is there enough detail provided in the methods for the work to be reproduced?

Yes.

Other minor points:

- (Fig. 2. caption) Typo. $f(1 - (1 - f^2)) \rightarrow f(1 - (1 - f)^2) = 2f^2 - f^3$

- The authors claim in Section "Effects of heterogeneous activity" that the heterogeneity in activity distribution increases the efficacy of contact tracing. Why they are effective around $v = 1.5$, not around the most heterogeneous case?

Reply to Reviewer #1

The article compares manual contact tracing with the approach based on exposure notification apps. The core message is that if an average probability of tracing a contact is assumed to be the same between manual and digital approach - even considering the delays inherent in the manual method - manual tracing is more efficient. The main reason behind this effect is that in the digital approach it is pre-determined who can and cannot be identified as a contact (by the virtue of having or not having a notification app installed). A super-spreader without an app will not be identified digitally, but is more likely to be identified manually because of their many contacts.

The paper points out an important shortcoming of digital contact tracing - the potential to miss superspreaders. As we learn more about the dynamics of spread, with a minority of carriers infecting the vast majority of susceptible individuals, this shortcoming proves to be more important than initially thought. Just because of this observation, I believe the paper constitutes a very important voice in the discussion. The presented results indicating comparatively high efficiency of manual tracing even with long delays are surprising and worth noting, since the slow operation is one of the main weaknesses pointed out against manual tracing. The paper is timely, as many countries still have not deployed digital tracing in any form while still struggling with implementing manual tracing. It is also written very clearly, most of the assumptions are well argued (apart from assuming that individuals with symptoms isolate immediately, but I don't believe it changes the outcomes of the comparison), and further examined for robustness in supplementary information.

We thank the Reviewer for the time dedicated to our manuscript and the positive appreciation of our work.

I am not qualified to comment on the correctness of the results regarding the calculation of epidemic thresholds, but the methods appear to be derived from established papers. I can however comment on framing of the research question. The main problem with the paper is that it presents the two approaches to contact tracing as a dichotomous choice. However, there is an apparent consensus, including even the creators of the DP3T privacy preserving digital approach, that digital exposure notification systems should only supplement manual tracing, not replace it. Most countries have manual tracing systems in operation and their question is whether it makes sense to adopt an app-based system in addition to the traditional approach. If the authors were to make a truly important contribution to the current discussion, they should acknowledge this current situation, and perform the simulations when both systems are employed concurrently.

The Referee is right, we realized that the previous version of the manuscript presented the two types of contact tracing as alternative rather than complementary and this is not what we want to convey to readers.

In the revised version, we have now included a major extension of our work and we consider a hybrid protocol with manual and digital contact tracing acting at the same time. The

results are contained in a new section in the main manuscript (*Section I.H: Hybrid contact tracing protocols: manual plus digital CT*), where we present our results on the increase of the epidemic threshold in the combined regime. With this extension, we can now estimate the relative increment on the threshold when the digital protocol is progressively added to the manual one. The details of this extension are discussed in the Supplementary Information (*Section I.C: Hybrid CT*). The new analysis confirms that if no targeted correlations are introduced between activity and adoption (see point below), most part of the increase in the epidemic threshold is due to the manual protocol.

We have also modified the paper in several points in the introduction and in the conclusions to put forward the combined application of the two methods. In particular, we discuss in details the gain introduced by the hybrid (digital + manual) protocol over a purely manual one in the new *Section I.H* and we show our results in Fig. 6.

Finally, I recommend that the authors refer in their discussion (or at least consider, as I am not sure about the policy around citing to the initial reports showing that app adoption appears to be reversely correlated with activity: people who install the app, tend to be more health conscious, work from home, and employ a range of other means to slow down the pandemic [2, 3]. That effect speaks even more strongly to the authors' point.

We thank the Reviewer for pointing out these important references. We have added and discussed them in the revised version in the introduction and in the conclusion.

The referee raises the important problem of correlation between adoption rates and social behavior. As well known, these type of data are scarce due to the privacy-preserving protocols of the apps. It would be extremely useful to have detailed data on this point. However, our formalism is very general so what we can do is test different correlations of adoption versus activity of nodes within our modelling scheme. Following the Referee's suggestion we added a section in the Supplementary Information (*Section III.D: Correlation between probability of app adoption and individual activity*) where we discuss three extreme cases of correlation: the uncorrelated case in which the adoption rate is constant, an anticorrelated case, with only the less active, careful and health conscious individuals adopting the app as discussed in the papers suggested by the Referee [2, 3], and a "virtuous" adoption, that is an adoption positively correlated with the activity of the nodes. In the latter case very active people, that are potential superspreaders, are very likely to download the app. We show our results in Fig. 5(d) in the Supplementary Information. This last policy is surprisingly effective in increasing the epidemic threshold, confirming the relevance of superspreaders in the propagation. More realistic shapes of the adoption $f(a_S)$ can be considered as interesting directions for future work.

I believe that pointing out the super-spreader weakness of digital contact tracing is crucial and the paper deserves publishing mostly as is. However, I would recommend that the authors rephrase the manuscript so as it does not present the choice between the two approaches as either or. Ideally, the authors would present the results of simulating both systems at work simultaneously, but the paper can also be published without it.

We totally agree with the Reviewer on the importance of studying hybrid CT approaches. As explained above, the revised version has been deeply modified and, after analysing separately the two protocols, it has as a central result the study of the effects of the combined protocol, manual plus digital.

[1] Digital Proximity Tracing in the COVID-19 Pandemic on Empirical Contact Networks, G. Cencetti et al (2020) [2] Are COVID-19 proximity tracing apps working under real-world conditions? Indicator development and assessment of drivers for app (non-)use, V. Wyl et al (2020) [3] Predicting public take-up of digital contact tracing during the COVID-19 crisis: Results of a national survey, Saw et al (2020)

Reply to Reviewer #2

The article presents an extremely policy relevant study with fascinating results. While I commend the effort, I have a number of remarks that I hope the authors could address to clarify the scope and applicability (and possibly validity) of their study.

We thank the Reviewer for her/his overall positive assessment of the manuscript.

The authors indicate realism as one of the motivations of their work, my main concern is with some of the assumptions of the study, which may end up restricting its realism and applicability.

A crucial implementation assumption when comparing the two CT strategies seems to be that all contact tracing relies exclusively on the strategy tested. While it is realistic to assume that, in the manual contact tracing (MCT) scenario, all tracing relies on recollection on the part of the individual, it is not as realistic to assume that all contact tracing in the digital contact tracing (DCT) scenario relies to the adoption of the app. E.g: if I test positive to the virus, my wife and children will know and isolate, even if they didn't install the app. It would be beneficial to the realism (and policy applicability) of the study if the authors assumed that a certain proportion (or a fixed number) of an individual's contacts are always traced, regardless of CT strategy, assuming that contacts within the household, or otherwise very proximal, can be traced without the need for a formal CT process.

We agree with the Reviewer about this very interesting remark.

In the revised version we have now considered a major extension of our model to include the hybrid protocol, that is manual + digital contact tracing, acting and sampling the population at the same time. This major extension and the corresponding results are discussed in a new section in the main manuscript (*Section I.H: Hybrid contact tracing protocols: manual plus digital CT*), where we derive the relative increment of the epidemic threshold when the digital protocol is added to the manual one. All the complex mathematical details of the hybrid protocol are discussed at pages 9, 10 and 11 of the Supplementary Information (*Section I.C: Hybrid CT*). The new analysis confirms that for realistic adoption rates and even considering delays and scalability, most part of the increase in the epidemic threshold is due to the manual protocol.

With the new hybrid analysis, we are now in the position to test the interesting effect mentioned by the Referee: the digital contact tracing is not only digital but it comes with a "built in" fraction of deterministic household manual tracing, performed directly by infected individuals on their stricter contacts (family, close friends). In the Supplementary Information we have added a new section (*Section III.C: Deterministic Household CT*), devoted to the study of contact tracing protocols where a fixed number of contacts is always traced manually, both in digital and in manual contact tracing. In the digital case we assume that the index case manually traces at least s contacts (household size) or, if she has had less, she traces them all.

In Fig. 5(a)-(c) of the Supplementary Information we show our results of this augmented protocol. We show that the manual contact tracing is more advantageous than the digital one even in this case. At realistic values of adoption rates, most part of the increase in the epidemic threshold is always due to the manual protocol. The digital protocol becomes more effective only for very high app adoption rates and unrealistically long delays: therefore, our results are robust with respect to addition of this household CT term.

Certain assumptions around the number of contacts being traced seem to implicitly favour the manual strategy. In particular, a maximum of $k_C = 130$ seems overwhelmingly large, even according to the literature that the authors themselves refer to. Ref.52 (page 6) mentions 10-30 as the interval of contacts normally manually traceable for each individual. The values tested in the sensitivity analysis also appear to overestimate this number. Traceable or not, 130 contacts might be an extreme case for an individual with high a and b in the power-law network. I'd like the authors to add some detail on the contact patterns emerging from their model. A simple plot of the distribution of contacts and/or infection events could be useful to assess the realism of the patterns generated.

We agree with the referee that this point has to be clearly discussed. In the Supplementary Information we present our results for smaller values of $k_C = 65$, obtaining results comparable with those presented in the main manuscript.

However, we point out that $k_C = 130$ is the maximum number of traced contacts, while the average number is 10 – 60, depending on the value of the recall probability $\varepsilon = 0.1 - 0.5$ (the contacts are always the cumulative contacts in the 14 previous days). We have used these values because they match the analysis and in particular Fig. 2(b) in Ref. 47 of the revised manuscript, where data for the number of traced contacts are discussed. Cases of traced contacts of order 100 and even more are frequently observed and they are related to tracing protocols in large communities (schools, hospitals, celebrations). As requested by the Referee, in the Supplementary Information we now plot in Fig. 3(a) the distribution of the number of contacts traced by an index case with manual tracing in the presence of limited scalability with $k_C = 130$ and different values of $\bar{\varepsilon}$. This plot confirms that our values are consistent with the data reported in Ref.47. In the same figure, we also plot now the distribution of the number of contacts. Note that these numbers always refer to the cumulative number of contacts in the previous time window, that is 14 days here.

The limited scalability assumption also seems very lenient on the MCT strategy. The only scalability issue implemented is a limitation to the recollection ability of individuals. However we know that, unfortunately, when the infection rate becomes high enough, MCT systems with limited manpower are likely to collapse, resulting in a failure to interview a proportion of infected agents altogether. DCT should (at least in principle) escape this further limitation as they are less reliant on manpower. I believe that, in a revised version of the paper, the authors should, at least, add this remark to the limitations of the model. Or they could try and implement a more realistic scalability limitation by not only assuming a lower k_C , but also that some nodes may not return any contact (or very few, just the closest) to the MCT system, especially at the peak of the epidemic.

We thank the referee for the very pertinent remark.

We have now clarified in the revised paper that our analysis and our comparison of digital and manual contact tracing applies in the early stages of the epidemic, when the number of infected people is sufficiently low so that manual contact tracing is practically doable and not excessively costly. In later stages of the evolution, manual CT may fail, as currently observed for example in Italy. At that point, it is clear that the whole tracing procedure is extremely less effective in containing the epidemic and different measures, as massive social distancing, have to be taken to limit the spreading. Our results can be relevant before we arrive at that stage.

One final remark. In the Introduction the authors seem to imply that DCT is an alternative to MCT. This is not the case: where DCT is implemented it is alongside MCT, the two strategies are seen as complementary, not alternative. After all, as the authors correctly state, generally around 50% of contacts can be easily traced. DCT exists for the other 50%.

This is a very important remark. We have now modified the paper to correct this unwanted message that the previous version conveyed. As already discussed above, the revised version has, as a central result, the interplay of the two protocols and the hybrid approaches. The introduction and the conclusions have been changed accordingly.

There's a mistype in the Fig.2 caption, where the exponent of the first term of the last equation should be outside the parenthesis. The calculation is correct nevertheless.

We thank the Reviewer for pointing out the mistake, that we have corrected.

Reply to Reviewer #3

The authors investigated the effectiveness of manual and digital contact tracing using an activity-driven model with parameters tailored to COVID-19. The authors showed that, although digital contact tracing can be performed instantaneously at scales, digital contact tracing is less effective than manual contact tracing. This is because digital contact tracing cannot trace infections from or to people without a tracing app while manual tracing can, calling a necessity of manual contact tracing to mitigate the current pandemic. The manuscript is well written and clear in most of its passages, and I have some minor comments and questions for the authors which I think can improve readability.

- What are the noteworthy results?

Manual contact tracing can be more effective than digital contact tracing even when digital contact tracing is performed instantaneously at scales.

- Will the work be of significance to the field and related fields? How does it compare to the established literature? If the work is not original, please provide relevant references. Yes. The authors clearly position their work in the literature. Specifically, related studies investigated the effectiveness of digital contact tracing based on simulations and case reports, calling that app adoption rate plays a crucial role. Essentially, the authors approach the same question but take a different approach, where they compare the effectiveness of manual and digital contact tracing.

We thank the Reviewer for her/his positive appreciation of our work.

- Does the work support the conclusions and claims, or is additional evidence needed? One issue is that they assume that manual tracing can trace at most k_c contacts per each infected individual. This is indeed not true for manual contact tracing; as the number of new cases continues to grow, one may be compelled to spare less time and resources per individual. Therefore, the number of traceable contacts per individual would decrease as the new cases grow. Therefore, their claim is limited to the early stage of epidemic spreading, which is neither discussed nor stated in the manuscript.

We agree with the remark of the Referee.

We have clarified this point in the revised version. Indeed, as correctly stated by the Referee, our analysis applies in the early stages of the epidemic, when the number of infected people is sufficiently low so that manual contact tracing is practically doable and not excessively costly. We have now stressed more this limitation in the revised paper.

- Are there any flaws in the data analysis, interpretation, and conclusions? - Do these prohibit the publication or require revision?

No.

- Is the methodology sound? Does the work meet the expected standards in your field? I suggest the authors to show the variations (e.g., standard error or confidence interval) in the figures.

We thank the referee for this remark.

The main results of our work concern the epidemic threshold, which is obtained analytically: for this reason, most of the results, being analytical, do not feature error or confidence interval (Fig. 1-4 of the manuscript). On the other hand, the numerical results regarding the epidemic active phase feature errors that are smaller or comparable with the thickness of the curves, therefore we do not show the error bars on the plot (Fig. 5 of the manuscript).

In the revised version of the manuscript, we clarify this aspect by adding a sentence in the caption of Fig. 5 (numerical simulations of the epidemic active phase), indicating the number of realizations on which the curves are averaged and commenting on the error.

- Is there enough detail provided in the methods for the work to be reproduced?
Yes.

Other minor points:

- (Fig. 2. caption) Typo.

$$f(1 - (1 - f^2)) \rightarrow f(1 - (1 - f)^2) = 2f^2 - f^3$$

Thank you, we have corrected this mistake.

- The authors claim in Section "Effects of heterogeneous activity" that the heterogeneity in activity distribution increases the efficacy of contact tracing. Why they are effective around $\nu = 1.5$, not around the most heterogeneous case?

We thank the referee for asking this question on this subtle and important point.

The heterogeneity in the activity, that is the presence of superspreaders, always enhances the effectiveness of the two protocols. The exact value of the parameter ν at which the effectiveness is maximum depends on the value of the other parameters appearing in the model. In particular, if the power law distributed activity has an upper and a lower cut-off with a fixed ratio (in our case $a_m < a < 10^3 a_m$) and if one works at fixed average activity $\langle a \rangle$, then the maximum heterogeneity of the distribution, for example measured by looking at $\langle a^2 \rangle / \langle a \rangle^2$, is always at $\nu = 1$. The value of the epidemic threshold, calculated for example in Eq. 25 and Eq. 63 of the Supplementary Information, do not depend solely on activity fluctuations, i.e. $\langle a^2 \rangle$, but also on higher moments of the activity a , as $\langle a^3 \rangle$, so the maximum effectiveness is observed at higher values of ν , always in the region $1 < \nu < 2$, and it also depends on other parameters of the models (see Eq. 25 and Eq. 63 in the Supplementary Information). To clarify this subtle point, we have added a short paragraph in the Supplementary Information (*Section I.D.4: Heterogeneous case*) and a sentence in the main manuscript in *Section I.D: Effects of heterogeneous activity*.

Summary of changes

All changes with respect to the previous version are marked in red and are detailed in the following summary.

Changes made to the main text:

- *Page 1*: we slightly changed some words and sentences in the abstract and introduction, to better convey the message that manual and digital CT protocols are complementary;
- *Page 2*: in the middle and at the end of the introduction we added two sentences, to better convey the message that the two CT protocols are complementary and to summarize our new results on hybrid CT protocol;
- *Page 4*: in the caption of Fig. 2 we corrected the mistype;
- *Page 4*: at the end of the section *I.B.1 Manual CT* we added a paragraph which clarifies that our analysis, focusing on the epidemic threshold, applies in the early stages of the epidemic;
- *Page 4*: at the end of the section *I.B.2 App-based CT* we added a sentence describing the hybrid CT protocol;
- *Page 5*: in section *I.C Stochastic vs. prearranged sampling* and in section *I.D Effects of heterogeneous activity* we changed some sentences to make the sections more clear and we added a sentence to clarify the conditions for the activity distribution to be maximally heterogeneous, also commenting the dependence of the epidemic threshold on the activity distribution moments;
- *Page 7*: we added a sentence at the end of caption of Fig. 5, indicating the number of realizations on which the curves are averaged and commenting on the error;
- *Page 7*: in section *I.G Robustness* we added a paragraph with a description of the new robustness analysis related to the deterministic household CT;
- *Page 7*: we added the section *I.H Hybrid contact tracing protocols: manual plus digital CT* in which we describe our results regarding the effectiveness of hybrid CT protocol and the relative performance of manual and digital CT in this combined procedure;
- *Page 8*: we added a new figure (Fig. 6) in which the results regarding the implementation of hybrid CT protocol are shown;
- *Page 8*: we reformulated the last part of section *II Discussion*, discussing the new hybrid CT protocol effects, the role of correlations in app adoption and future perspectives regarding these results;
- *Page 9*: at the end of section *III.A Mean-field equations and implementation of CT protocols* and section *III.B Epidemic thresholds* we added two short sentences which describe methods also for the hybrid CT protocol;

- *Page 10*: at the end of section *III.C Model parameters* we modified a sentence and added a new one, to clarify the meaning of k_C ;
- *Page 10*: we added the *Data availability* and *Code availability* statements, also rearranging the sections order accordingly to editorial requirements;
- *Page 12*: we added the Refs. [51,52,55], which are cited in the revised version.

Changes made to the Supplementary Information (SI):

- *Page 1*: we added the *Table of Contents* to clarify the structure of the SI and make its content more clear to the reader;
- *Page 1*: we added a sentence at the end of the short introduction, describing the new contents of the SI;
- *Page 2*: at the beginning of section *I.A Manual CT* we added a paragraph which describes in further detail the epidemic model with manual CT, explicitly writing the possible transitions between compartments. Moreover, we added a new figure (Fig. 1) with a scheme of the epidemic model in the presence of CT;
- *Page 5*: at the beginning of section *I.B Digital CT* we added a paragraph which describes in further detail the epidemic model with digital CT, explicitly writing the possible transitions between compartments;
- *Page 9-11*: we added the section *I.C Hybrid CT*, in which we describe the hybrid CT protocol. This section contains the mean-field equations and the analytical derivation of the epidemic threshold in the presence of the hybrid CT protocol;
- *Page 12-13*: we added the section *I.D.4 Heterogeneous case*, in which we consider the activity distribution adopted for the realistic heterogeneous case and we analyze the distribution heterogeneity and fluctuations, also by adding a new figure (Fig. 2);
- *Page 14*: we divided section *III Robustness of the results* into subsections. The section *III.A Activity-attractiveness distribution, limited scalability parameters and delays* collects the content already present in the previous version of the SI regarding the robustness of the results as far as limited scalability and delays are concerned (results shown in Fig. 3(b-d)). Section *III.B Epidemic active phase* collects the content already present in the previous version of the SI related to the robustness of the results as far as the epidemic active phase is concerned (results shown in Fig. 4(a-d)). We added some sentences to caption of Fig. 4, indicating the number of realizations on which the curves are averaged and commenting on the error;
- *Page 14-15*: we added the new panel (a) of Fig. 3, with the contacts distributions. Furthermore, at the beginning of section *III.A Activity-attractiveness distribution, limited scalability parameters and delays* we added a paragraph describing the new panel and the contacts distributions;

- *Page 14-17*: we added a new figure (Fig. 5) and a new section *III.C Deterministic household CT*, which describes the results related to the deterministic CT on household, shown in Fig. 5(a-c);
- *Page 16-17*: we added a new section *III.D Correlation between probability of app adoption and individual activity*, which describes the results on the role of the correlations between app adoption and node activity, shown in Fig. 5(d);
- *Page 18*: we added Refs. [7-10], which are cited in the revised version of SI.

Reviewer #1 (Remarks to the Author):

I appreciate how seriously and thoroughly the authors took the feedback me and the other reviewers. My reservations are satisfied with the changes they introduced. I look forward to the publication of this paper.

Reviewer #2 (Remarks to the Author):

Thank you for addressing my remarks, I deem the article ready for publication.

Reviewer #3 (Remarks to the Author):

I have the impression that the authors carefully addressed the comments. In my opinion, the paper can be published.

Reply to Reviewer #1

I appreciate how seriously and thoroughly the authors took the feedback me and the other reviewers. My reservations are satisfied with the changes they introduced. I look forward to the publication of this paper.

Reply to Reviewer #2

Thank you for addressing my remarks, I deem the article ready for publication.

Reply to Reviewer #3

I have the impression that the authors carefully addressed the comments. In my opinion, the paper can be published.

We thank the Reviewers for the time dedicated to our manuscript and for the positive appreciation of our work. Their careful reports and questions helped us to improve the manuscript a lot.